# Effects of Light Intensity on the Growth and Biochemical Composition in Various Microalgae Grown at High CO_2_ Concentrations

**DOI:** 10.3390/plants12223876

**Published:** 2023-11-16

**Authors:** Elizaveta A. Chunzhuk, Anatoly V. Grigorenko, Sophia V. Kiseleva, Nadezhda I. Chernova, Mikhail S. Vlaskin, Kirill G. Ryndin, Aleksey V. Butyrin, Grayr N. Ambaryan, Aleksandr O. Dudoladov

**Affiliations:** 1Joint Institute for High Temperatures of the Russian Academy of Sciences, 125412 Moscow, Russia; presley1@mail.ru (A.V.G.); k_sophia_v@mail.ru (S.V.K.); chernova_nadegda@mail.ru (N.I.C.); kirillryndin2011@gmail.com (K.G.R.); aleksey.butyrin@yandex.ru (A.V.B.); ambaryan1991@gmail.com (G.N.A.); nerfangorn@gmail.com (A.O.D.); 2Faculty of Geography, Lomonosov Moscow State University, 119991 Moscow, Russia

**Keywords:** high CO_2_ concentrations, light intensity, microalgae, viability of microalgae cells

## Abstract

In modern energy, various technologies for absorbing carbon dioxide from the atmosphere are being considered, including photosynthetic microalgae. An important task is to obtain maximum productivity at high concentrations of CO_2_ in gas–air mixtures. In this regard, the aim of the investigation is to study the effect of light intensity on the biomass growth and biochemical composition of five different microalgae strains: *Arthrospira platensis*, *Chlorella ellipsoidea*, *Chlorella vulgaris*, *Gloeotila pulchra*, and *Elliptochloris subsphaerica*. To assess the viability of microalgae cells, the method of cytochemical staining with methylene blue, which enables identifying dead cells during microscopy, was used. The microalgae were cultivated at 6% CO_2_ and five different intensities: 80, 120, 160, 200, and 245 μmol quanta·m^−2^·s^−1^. The maximum growth rate among all strains was obtained for *C. vulgaris* (0.78 g·L^−1^·d^−1^) at an illumination intensity of 245 µmol quanta·m^−2^·s^−1^. For *E. subsphaerica* and *A. platensis*, similar results (approximately 0.59 and 0.25 g·L^−1^·d^−1^ for each strain) were obtained at an illumination intensity of 160 and 245 µmol quanta·m^−2^·s^−1^. A decrease in protein content with an increase in illumination was noted for *C. vulgaris* (from 61.0 to 46.6%) and *A. platensis* (from 43.8 to 33.6%), and a slight increase in lipid content was shown by *A. platensis* (from 17.8 to 21.4%). The possibility of increasing microalgae biomass productivity by increasing illumination has been demonstrated. This result can also be considered as showing potential for enhanced lipid microalgae production for biodiesel applications.

## 1. Introduction

The development of modern energy is facing various challenges, including the regional limitation of fossil fuel reserves [1] and the complex impact of the energy production process on the environment. In recent years, the task of reducing greenhouse gas emissions [2] has acquired a special place in connection with climate change trends. Several technologies have been proposed and tested to prevent CO_2_ emissions and capture them from the atmosphere. One of the widely studied and highly effective methods of CO_2_ capture is to use microalgae as photosynthetic organisms that absorb CO_2_ and can be used for biofuel production [3,4,5]. The advantages of microalgae as photosynthetic agents of carbon conversion have been repeatedly discussed. It is worth emphasizing the most important of them: (1) the cultivation of microalgae does not require arable land, unlike terrestrial crops, and the use of wastewater as a nutrient medium reduces the consumption of water and inorganic components of nutrient media [6,7,8]; (2) microalgae can synthesize valuable metabolites, vitamins, and various organic compounds in commercially significant volumes [9,10]. For effective CO_2_ absorption, it is necessary to select the most productive strains of microalgae tolerant to high concentrations of CO_2_ and flue gases [11,12,13], as well as to search for optimal cultivation conditions to obtain maximum biomass productivity. A wide range of microalgae is used as objects of research, among which the most common are strains of *Chlorella*, *Nannochloropsis*, in some cases *Arthrospira* (*Spirulina*). The results obtained on the productivity of microalgae biomass vary in a wide range, apparently due to heterogeneous cultivation conditions (type of photobioreactor, temperature, method of supplying gas–air mixtures, gas consumption during bubbling, duration of cultivation, etc.).

Generalization and comparative analysis of the results of experiments with increased content of CO_2_ in the gas–air mixture (10 and 20%) are given, in particular in [14], with the absorption rates of CO_2_ (from 0.160 to 0.265 g·L^−1^·d^−1^, depending on the strains and concentration of CO_2_), and the achieved content of lipids, chlorophyll, and carotenoids. A similar generalization can be seen in Ho et al. [15]. Note that both reviews do not indicate illumination levels during cultivation. In their article, Hu Xia et al. [16] studied the response of ten *Chlorella* strains to increased CO_2_ concentrations, and the maximum productivity for four of them was at 10% CO_2_; *C. vulgaris* was identified, demonstrating the maximum accumulation of lipids. It is necessary to note that the cells of both *Chlorella* sp. and *C. vulgaris* kept their normal morphologies after 15-day batch culture, while the cells of two other strains were destroyed, which was confirmed by photographs from a scanning electron microscope. This confirms the legitimacy of choosing a method of long-term microalgae cultivation for the most complete characterization of their growth in the face of elevated CO_2_ concentrations and optimization of other cultivation conditions. In several studies, *A. platensis* is used as a convenient object due to the resistance to contamination and the manufacturability of growing and harvesting biomass. Thus, when growing the *A. platensis* strain in gas–air mixtures with CO_2_ concentrations of 1, 5, and 9%, an average biomass growth rate (duration of experiments 15 days) of 0.079, 0.076, and 0.048 g·L^−1^·d^−1^, respectively, was achieved [17]. Adaptation of this strain to elevated concentrations of CO_2_ allowed further experiments on its cultivation in a gas–air mixture simulating flue gases with a CO_2_ content of 6% [18], to obtain a growth rate almost twice as high, 0.140 g·L^−1^·d^−1^ (the duration of the experiments is 14 days). The task of studying the effect of illumination on the growth of microalgae was set earlier and studied in detail. The study of the effect of illumination showed that, as a rule, with an increase in light intensity, the growth rate of microalgae increases until the photoinhibition threshold is reached, but both the result of this effect and the threshold are different in different strains [7,19,20]. Thus, the influence of different levels of illumination as well as the duration of periods of illumination (photoperiods) on the growth of a wide range of microalgae strains (*Porphyridium purpureum*, *Chloromonas reticulata*, *Parietochloris incisa*, *Neochloris*, *Botryococcus braunii*, *Scenedesmus obliquus*, *A. platensis*, *Desmodesmus* sp., etc.) were considered [7,19,21,22,23,24,25,26]. The most interesting are the study results of the effect of illumination when cultivating microalgae at high CO_2_ concentrations. Thus, the reaction of *Euglena gracilis* to variations in temperature (25–33 °C), CO_2_ concentration (from 2 to 6%), and illumination (20–200 µmol quanta·m^−2^·s^−1^) were experimentally shown [27]. The maximum productivity values (approximately 0.045 h^−1^) were achieved at 4% CO_2_ and an illumination intensity of 100 µmol quanta·m^−2^·s^−1^. From those considered in [14], oleaginous microalgae marine *Chlorella* sp., freshwater *Chlorella* sp., *Scenedesmus* sp., *Botryococcus* sp., and *Nannochloropsis* sp., grown at CO_2_ concentrations in flue gases of 10 and 20%, the maximum values of biomass growth were obtained at 10% CO_2_ and an illumination intensity of 60 µmol quanta·m^−2^·s^−1^. The highest lipid content was shown by the strain *Nannochloropsis* sp.; and with the transition from the 12:12 photoperiod to continuous illumination, the growth rates for biomass and lipids increased. A further increase in the concentration of CO_2_ led to a decrease in productivity both in biomass and lipids. In their article, when Shih-Hsin Ho et al. [15] studied *Scenedesmus obliquus CNW-N*, a productivity of 0.84 g·L^−1^·d^−1^ was achieved for the this strain at a much higher illumination intensity of 420 µmol quanta·m^−2^·s^−1^ (temperature 28 °C, 2.5% CO_2_, illumination constant; the range of studied illumination intensity is 60–540 µmol quanta·m^−2^·s^−1^). Strain *Chlorella* sp. *MTF-7* was grown in a flue gas atmosphere at CO_2_ concentrations of 2%, 10%, and 25% at a constant illumination intensity of 300 µmol quanta·m^−2^·s^−1^ [28]. With such a significant range of CO_2_ concentrations, the achieved biomass productivity was approximately the same in all experiments, and it was approximately 0.350 g·L^−1^·d^−1^. It should be noted that both the high level of illumination and CO_2_ concentrations did not lead to inhibition of the growth of *Chlorella* microalgae and provided high biomass productivity.

The search for optimal conditions is also conducted in the direction of the spectral composition of light used for microalgae cultivation [24,26,29,30,31] and the optimal duration of the photoperiod [25,32,33]. The influence of the spectrum is outside the topic of this work, although it is worth noting that several publications have shown that the daylight spectrum as a whole is close to monochrome lighting in terms of its effect on the productivity of microalgae. Some studies have found a decrease in the rate of carbon fixation in the presence of even a short dark period compared to cultures under continuous illumination [25,34]. This conclusion determined the illumination modes we selected. If LED lighting is used, this does not significantly increase the cost of cultivation. Since the absorption of CO_2_, as a method of reducing emissions into the atmosphere, is associated with the further transformation of microalgae biomass into energy products [35], the criterion for optimizing the cultivation of microalgae, in addition to the rate of biomass growth and the intensity of CO_2_ absorption, is the content of the main biochemical components, especially lipids as raw materials for the production of biodiesel [14,21,22,29,32]. It is important to note that high CO_2_ concentrations are a stress for microalgae, and therefore various adaptation methods are used in experimental studies, in particular the Adaptive Laboratory Evolution (ALE) technique described in detail in [11]. ALE is an innovative stress-induced approach, the main advantage of which is that it can improve the characteristics of the strain and develop the resistance of microalgae to numerous environmental stresses, supporting rapid cell growth [35,36]. The ALE technique was successfully used [37,38,39].

In our previous studies [11], as a result of long-term phased cultivation with adaptation to increasingly high CO_2_ concentrations, as indicated above, stable and high productivity of microalgae *Arthrospira platensis*, *Chlorella ellipsoidea*, *Chlorella vulgaris*, *Gloeotila pulchra*, and *Elliptochloris subsphaerica* was achieved (CO_2_ up to 9% in a gas–air mixture). However, microalgae cultivation in these experiments was carried out at low light values (74.3 µmol quanta·m^−2^·s^−1^). In this regard, the purpose of these studies is to determine the optimal illumination for strains adapted to high concentrations of CO_2_ by the criterion of productivity and the growth rate of biomass, which determine the efficiency of CO_2_ absorption. Towards this goal, the following work has consistently carried out: optimization of cultivation conditions for microalgae strains previously adapted to elevated CO_2_ concentrations by gradually adapting them to a high illumination intensity using the method of long-term continuous cultivation; comparative analysis of the response of the most crucial characteristics of biomass growth (growth rate, biochemical composition, absorption intensity of the main components of nutrient media, morphological characteristics of microalgae) to the process of adaptation to increased illumination; evaluation of the viability of microalgae under various conditions based on the express method of cytochemical staining of cells with a lifetime dye with their subsequent control by light microscopy.

## 2. Results and Discussion

### 2.1. The Biomass Growth Rate and pH of the Culture Medium

Figure 1 shows the results of determining the growth rate of microalgae biomass for all conducted experiments with different lighting intensities.

*C. vulgaris* has demonstrated a tendency to increase the growth rate with an increase in illumination intensity from 80 to 120 µmol quanta·m^−2^·s^−1^. Further, at an illumination intensity of 120, 160, and 200 µmol quanta·m^−2^·s^−1^, the biomass growth rate was the same within the error limits (0.59 g·L^−1^·d^−1^); however, at 245 µmol quanta·m^−2^·s^−1^, an active increase in biomass occurred, at which the maximum value of the biomass growth rate among all strains was obtained—0.78 g·L^−1^·d^−1^ (Figure 1). In *C. ellipsoidea*, a steady increase in biomass was observed, positively correlating with the intensity of illumination. With an increase in the illumination intensity from 80 to 200 µmol quanta·m^−2^·s^−1^, there is a gradual increase in the growth rate. At an illumination intensity of 200 and 240 µmol quanta·m^−2^·s^−1^, the growth rate is constant within the error limits, with the maximum value being 0.72 g·L^−1^·d^−1^ (Figure 1). In the case of *G. pulchra*, an increase in intensity from 80 to 160 µmol quanta·m^−2^·s^−1^ led to an increase in the biomass growth rate; at 160 and 200 µmol quanta·m^−2^·s^−1^, it is approximately equal; the maximum growth rate is achieved with an illumination intensity of 245 µmol quanta·m^−2^·s^−1^ and is 0.47 g·L^−1^·d^−1^ (Figure 1). For *E. subsphaerica* and *A. platensis* strains, an increase in the illumination intensity above 160 µmol quanta·m^−2^·s^−1^ did not lead to a significant increase in the biomass growth rate, which was approximately 0.59 and 0.25 g·L^−1^·d^−1^, respectively, while the initial growth rates at 80 µmol quanta·m^−2^·s^−1^ were 0.32 and 0.13 g·L^−1^·d^−1^. Thus, at a light intensity of 160 µmol quanta·m^−2^·s^−1^, the same biomass productivity can be obtained as at 245 µmol quanta·m^−2^·s^−1^, which may indicate saturation for these strains.

Table 1 presents the results of biomass productivity obtained in previous studies of the growth of *C. vulgaris* and *A. platensis* as the most frequently considered CO_2_ absorption problems. It can be seen that it was possible to achieve higher biomass productivity with an increase in illumination intensity. Note that during all the experiments lasting 14 days each, there was no mass death of microalgae, and all microalgae culture showed biomass gain. Thus, there was no photoinhibition of the strains; and in the case of *E. subsphaerica* and *A. platensis*, saturation was reached at 160 µmol quanta·m^−2^·s^−1^ illumination. For strains of *C. vulgaris*, *C. ellipsoidea*, and *G. pulchra*, it would be advisable to conduct additional studies with a further increase in illumination.

In the case of *Chlorella* strains, a significant pH increase was observed in all experiments (Table 2). At the same time, many works [44,45,46] show that, when cultivating microalgae with bubbling air with high CO_2_ concentrations, pH decreases, that is, acidification. Our results might be explained by the fact that *Chlorella* was grown on a Tamiya nutrient medium, which is unstable, since during the growth of microalgae, the physiological absorption of NO_3_^−^ anions prevails compared to Na^+^ cations, which accumulate in the nutrient medium, which leads to an increase in pH.

For *A. platensis*, the pH values in the experiments showed a tendency to a slight increase (Table 2), which confirms the high buffering of the Zarrouk nutrient medium. Acidification during CO_2_ bubbling by H^+^ and HCO_3_^−^ ions is leveled by selective absorption of the HCO_3_^−^ anion and accumulation of the Na^+^ cation from NaHCO_3_ baking soda from the Zarrouk medium, but, in general, the addition of ions with acidic properties to the medium during CO_2_ bubbling prevents strong alkalinization of the nutrient medium during its long-term cultivation. For *E. subsphaerica*, the pH remained constant for three experiments, but, in the experiment with an illumination intensity of 200 µmol quanta·m^−2^·s^−1^, the pH decreased markedly. A similar trend for pH values was obtained for *G. pulchra*: at 120 µmol quanta·m^−2^·s^−1^, the pH is constant, and at 160 and 200 µmol quanta·m^−2^·s^−1^, acidification of the medium was noted (Table 2). As described earlier, such a decrease in pH is characteristic when microalgae are grown in an atmosphere with elevated concentrations of CO_2_ or flue gases. At the same time, no acidification of pH was observed in the experiment with an illumination intensity of 245 µmol quanta·m^−2^·s^−1^.

### 2.2. The Biochemical Composition of Microalgae Biomass

The effect of light on the content of proteins, lipids, and carbohydrates in five different microalgae strains was studied. With increasing illumination, the protein content (Figure 2a) showed multidirectional dynamics. For the *C. vulgaris* strain, a drop in protein content was noted (from 61.0 to 46.4%) with an increase in illumination intensity from 120 to 160 µmol quanta·m^−2^·s^−1^; further, at an illumination intensity of 160, 200, and 245 µmol quanta·m^−2^·s^−1^, the number of proteins remained at one level within the margin of error. For strains *C. ellipsoidea* and *E. subsphaerica*, the content of proteins did not change significantly during all experiments. In the case of *G. pulchra*, at illumination intensities of 120 and 160 µmol quanta·m^−2^·s^−1^, the protein content is the same within the error limits, with an increase in illumination from 160 to 200 µmol quanta·m^−2^·s^−1^, an increase in protein content is observed (from 32.6 to 46.5%), and at 200 and 245 µmol quanta·m^−2^·s^−1^ protein content again does not change significantly. A similar picture develops for strain *A. platensis*: at first, the protein content increases with an increase in illumination intensity from 43.8 to 57.2% (from 120 to 160 µmol quanta·m^−2^·s^−1^, respectively); with increasing intensity from 160 to 200, there is a noticeable decrease in protein content (from 57.2 to 31.9%), and no further change occurs (at 245 µmol quanta·m^−2^·s^−1^). Thus, a pronounced trend, namely, a decrease in protein content with an increase in illumination, was noted only in *C. vulgaris* and *A. platensis* strains. These results positively correlate with those obtained in [47], where there is also a decrease in the number of proteins with an increase in the intensity of illumination.

For lipids, as well as in the case of proteins, there was no unambiguous trend among all strains with an increase in the intensity of illumination. Let us note the main points on the results obtained (Figure 2b). For *C. vulgaris* and *C. ellipsoidea* strains, the maximum lipid content was recorded at 200 µmol quanta·m^−2^·s^−1^ (26.5 and 23.9%, respectively). In the margin of error for *E. subsphaerica*, similar results were obtained at illuminances of 120, 160, and 200 µmol quanta·m^−2^·s^−1^, which are 21.1, 20.7, and 20.8%, respectively. *G. pulchra* showed a tendency to decrease the content of lipids. For this strain, the maximum content was at an illumination intensity of 160 µmol quanta·m^−2^·s^−1^ (27.3%). At an illumination intensity of 245 µmol quanta·m^−2^·s^−1^, the lowest lipid content was recorded for all strains except *A. platensis*, while for this strain the maximum result was obtained. The results obtained for *A. platensis* (a slight increase in lipids (from 17.8 to 21.4%) with an increase in illumination from 120 to 245 µmol quanta·m^−2^·s^−1^) were consistent with the results in [48], where an increase in lipid content from 36.1 to 47.1% was noted with a change in illumination intensity from 55 to 400 µmol quanta·m^−2^·s^−1^. In the same work, the growth of lipids with an increase in illumination was also shown for the *C. vulgaris* strain (from 21.0 to 33.0% at 55 and 450 µmol quanta·m^−2^·s^−1^, respectively), but there is no such pronounced trend in our study for *Chlorella*.

Strains *C. vulgaris*, *C. ellipsoidea*, and *E. subsphaerica* were resistant to increased light intensity in terms of carbohydrate content (Figure 2c). In Nzayisenga et al. [7], a similar trend was noted for various strains of microalgae: while increasing illumination, the amount of carbohydrates does not change significantly. The carbohydrate content of *G. pulchra* strain remained unchanged in all experiments, except for an illumination intensity of 245 µmol quanta·m^−2^·s^−1^, at which the amount of carbohydrates increased significantly (25.6%). *A. platensis* shows a steady accumulation of carbohydrates in cells with increasing illumination intensity (from 9.9 to 23.1% at 120 and 245 µmol quanta·m^−2^·s^−1^, respectively).

### 2.3. Dynamics of the Nutrient Media Components during the Experiments 

The results of the consumption of nitrates and phosphates by microalgae strains are shown in Figure 3.

In *Chlorella* strains, the proportion of nitrate consumption from the initial amount (Figure 3a) lies in the range of 50–65% according to the results of all experiments with different lighting intensities. It can be said that due to the increase in illumination, there were no significant changes in the consumption of nitrates. In the case of three other strains (*E. subsphaerica*, *G. pulchra*, and *A. platensis*), the proportion of nitrate consumption varies from 94% to almost 100%, which could become a limiting factor for the further growth of microalgae biomass under continuous cultivation. According to the results of phosphate consumption (Figure 3b), no definite trend has been identified. A major portion of the values (approximately 80%) are from 20 to 60%. However, for *G. pulchra*, at an illumination intensity of 200 µmol quanta·m^−2^·s^−1^, the absorption of phosphates reaches almost 100%; and at 245 µmol quanta·m^−2^·s^−1^, decreases to 50%.

The content of bicarbonates in the medium showed stable growth during all experiments with different illumination intensities for strains *C. vulgaris*, *C. ellipsoidea*, *E. subsphaerica*, and *G. pulchra* (Figure 4a–d). Note that bicarbonates are the main elements of the Zarrouk medium (for *A. platensis*), and initially are not part of the other nutrient media (Tamiya and BG-11).

In the case of *A. platensis* strain, a decrease in HCO_3_^−^ is observed at illumination intensities of 120 and 245 µmol quanta·m^−2^·s^−1^, and at 160 and 200 µmol quanta·m^−2^·s^−1^; on the contrary, an increase in the content of bicarbonates.

### 2.4. The State of Microalgae Cells under the Influence of Different Illumination Intensities

Microscopy of strains grown in continuous culture at 120 µmol quanta·m^−2^·s^−1^ for 12 days showed that visually microalgae cells did not differ in morphometric parameters from culture cells from experiments with 80 µmol quanta·m^−2^·s^−1^. Microscopy results of microalgae strains stained with the lifetime dye methylene blue showed the absence or minimal number of dead cells. Only for *G. pulchra* on the last day of the experiment (120 µmol quanta·m^−2^·s^−1^), a small number of single cells formed as a result of the decay of individual filaments were observed, which were stained with methylene blue dye, dead culture cells (Figure 5e), which may be due to a change in growing conditions (increased illumination intensity). Examples of photos are presented in Figure 5, which demonstrate the absence of mass staining of cells, which means the integrity of the cell walls of microalgae. These photos were taken in conditions close to production conditions using a conventional light microscope, directly at the place of experiments, where an atmospheric gas chamber (AGC) with the photobioreactors (PBRs) were installed.

In this experiment (at 120 µmol quanta·m^−2^·s^−1^) with *A. platensis* and *G. pulchra* algae, the mucous membranes of microalgae trichomes are partially stained (Figure 5d,f), but not algae cells. The photo of *G. pulchra* shows the cells of *E. subsphaerica*, which, as a result of partial contamination, were detected in small quantities already in an experiment with an illumination intensity of 120 µmol quanta·m^−2^·s^−1^. Note that Figure 5d shows a cluster of stained *E. subsphaerica* cells, which indirectly indicates the displacement of this strain by the dominant strain *G. pulchra*. In the case of two strains, *Chlorella* and a strain of *E. subsphaerica*, in all experiments with different illumination, the minimum number of colored cells was noted, and the morphology of the cultures was not visually changed (Figure 6). That is, the microalgae cells remained alive during the experiments, which indicates that the viability of the cultures was preserved under the influence of increased illumination.

Microscopy of the strain *G. pulchra* grown at 160 µmol quanta·m^−2^·s^−1^ illumination shows that there were significant morphological changes in the cells of the culture grown for 6 days, compared with the results at 120 µmol quanta·m^−2^·s^−1^. Figure 7 shows that under the influence of illumination, there was a massive disintegration of *G. pulchra* filaments into individual cells. However, its subsequent cultivation with an increase in the illumination intensity to 200 µmol quanta·m^−2^·s^−1^ (Figure 7b) revealed that this culture adapted and returned to its original state with the formation of first short 2–4 cells, and then long filaments of cells surrounded by a dense mucous membrane with radial cilia. A further increase in the illumination intensity from 200 to 245 µmol quanta·m^−2^·s^−1^ did not lead to the mass decay of *G. pulchra* filaments (Figure 7c).

Figure 8 shows the trichomes of *A. platensis*. Very long trichomes and a small number of short trichomes are visible in the field of view, which indirectly indicates a weakening of the process of cell growth and division (Figure 8a–d). At an illumination intensity of 245 µmol quanta·m^−2^·s^−1^, the trichomes of this culture were found to have disintegrated into separate parts (Figure 8e), and these parts are completely colored with methylene blue. Under the influence of lighting, the shell of the cells was destroyed, which led to the release of their contents to the outside, which was also stained with dye.

## 3. Materials and Methods

### 3.1. Microalgae Strains

As a result of primary screening, microalgae strains *Chlorella ellipsoidea rsemsu Chl-el*, *Chlorella vulgaris rsemsu Chv-20/11-Ps*, *Elliptochloris subsphaerica rsemsu N-1/11-B*, *Gloeotila pulchra rsemsu Pz-6*, and a resistant consortium of microalgae/cyanobacteria *Arthrospira platensis rsemsu P Bios* with heterotrophic bacteria (heterotrophic bacteria are the representatives of the genera *Pseudomonas*, *Bacterium* and *Bacillus*) were selected from the collection of the Renewable Source Energy Laboratory at Lomonosov Moscow State University (RSE LMSU). The selected strains have a sufficiently high biomass productivity and are resistant to changes in environmental parameters. A detailed description of microalgae cultures and the composition of nutrient media for them, as well as the rationale for choosing the listed microalgae for conducting experiments with different illumination at high CO_2_ concentrations, are presented in earlier work [11] (pp. 12–13). Consequently, this description is omitted in this paper.

### 3.2. Illumination Intensity

Experiments with various microalgae were carried out with a gradual increase in the illumination intensity: 80, 120, 160, 200, and 245 μmol quanta·m^−2^·s^−1^. The lighting was constant—24 h per day. The Flux Apogee MQ-200 (Utah, USA) was used to measure the illumination intensity. The experimental technique, described above, was chosen to adapt microalgae to high light values. 

### 3.3. Experimental Setup

To conduct experiments on growing microalgae at different illumination intensities and high CO_2_ concentrations, a laboratory installation was created, which includes the following main elements: the atmospheric gas chamber (AGC) and photobioreactors (PBRs) (10 pcs). A detailed description and design of the AGC and PBR are presented in [11]. Figure 9 shows the general view of the PBR system inside the AGC and the appearance of the AGC.

### 3.4. Experimental Procedure

The experiments were carried out using the PBR in the amount of 10 pieces, placed in the AGC. Uniform illumination was provided around the perimeter and height of the PBR using strip LED lights. Illumination was constant (i.e., 24 h per day). The gas–air mixture was fed to the PBR through air aerators, while the gas flow rate adjusted by the compressor was 1 L·min^−1^. Each reactor was covered with a textile cover on top to minimize contamination of the PBR with nontarget microalgae strains.

Experiments were carried out on microalgae cultivation on gas–air mixtures with CO_2_ content of 6% and at different illumination intensities: 80, 120, 160, 200, and 245 μmol quanta·m^−2^·s^−1^. Each experiment was conducted according to the following algorithm: (1) preparation of nutrient medium on distilled water in a volume of 8 L for each strain, seeding the medium with an inoculum of each strain to the initial concentration of microalgae biomass (0.2–0.25) g·L^−1^. Filling of two PBR with culture liquid (nutrient medium with inoculum) at the rate of 4 L in each PBR for each strain; (2) placement of the PBRs in the AGC, turning on illumination to a given intensity and bubbling. CO_2_ injection to a concentration of 6% in the AGC, sealing of the chamber; (3) cultivation of microalgae during 12 days at given illumination intensity. After the end of the experiment, microalgae biomass with culture liquid was placed into 5 L containers for further use. This biomass was used as a source of inoculum for PBR seeding in the following experiment.

### 3.5. Research Methods

The conditions of the experiments were as follows. Microalgae strains: *A. platensis*, *C. ellipsoidea*, *C. vulgaris*, *E. subsphaerica*, *G. pulchra*. Duration: 12 days for each experiment. Distilled water was used for preparation of nutrient media. Concentration of CO_2_ in the AGC was 6%. Temperature in the AGC: 27 ± 1 °C. PBR’s illumination intensity in each experiment: 80, 120, 160, 200, and 245 μmol quanta·m^−2^·s^−1^. 

The schedule of sampling, analysis of the state of microalgae (biochemical analysis, optical density of biomass—OD, pH of the medium, and microscopy), as well as the composition of the culture medium are presented below: the composition of the culture medium—on days 0 and 12; microscopy—on days 0, 6, and 12; OD and pH—on days 0, 3, 6, 9, and 12; biochemical analysis—after the end of each experiment.

The following methods and measuring instruments were used to measure these characteristics: OD determination by using photometer Expert-003 (Russia), pH determination by using the pH meter Expert-pH (Russia), bicarbonates determination by using titration, nitrates and phosphates determination by using ICS-1600 ion chromatograph (California, USA) with conductivity detector.

Microscopic monitoring of the state of microalgae cultures and their viability was carried out using a Mikmed-5—LOMO (Russia) light microscope. A method of cytochemical staining with a lifetime dye methylene blue was used for determining the viability of microalgae cells, which revealed dead (stained) cells under microscopy. At least 10 fields from each sample of experiments taken on the 0th, 6th, and 12th day were viewed; at the maximum magnification of the microscope, the number of blue-stained (dead) cells or their clusters was counted, as well as a botanical description of the state of the cells and photography.

Microalgae extraction was further conducted following the Folch procedure to determine the amount of lipids [49]. After lipid extraction, the remaining sediment was dried in a Binder VD53 (Germany) drying oven at 50 °C for 20 h. The content of proteins was measured using the method in [50]. The biomass sediment remaining after the protein extraction was then dried in a Binder VD53 desiccator at 60 °C for 20 h. The phenol-sulfuric acid method was used to determine the amount of carbohydrates [51]. All samples were analyzed 2 times.

## 4. Conclusions

This study is devoted to determining the optimal illumination for microalgae strains previously adapted to high CO_2_ concentrations to increase the efficiency of CO_2_ absorption. To achieve these goals, experiments were conducted on the cultivation of microalgae strains (*A. platensis*, *C. ellipsoidea*, *C. vulgaris*, *E. subsphaerica*, and *G. pulchra*) at different intensities (80, 120, 160, 200, and 245 µmol quanta·m^−2^·s^−1^) and 6% CO_2_.

To optimize the cultivation conditions of microalgae strains and obtain a stable result in biomass productivity, a mode of sequential adaptation to high illumination, using the method of long-term continuous cultivation, was implemented. The effectiveness of this approach has been confirmed by previous studies [11], where an increase in microalgae biomass was observed under high CO_2_ content (up to 9%). The growth rate was different because of species and strain specificity: for *C. vulgaris*, *C. ellipsoidea*, and *G. pulchra* strains, the maximum growth rates were obtained at 245 µmol quanta·m^−2^·s^−1^ (0.78, 0.72 and 0.47 g·L^−1^·d^−1^, respectively); for *A. platensis* and *E. subsphaerica*, 0.25 and 0.59 g·L^−1^·d^−1^, respectively, at 160 µmol quanta·m^−2^·s^−1^. According to the results of biochemical analysis, it is also possible to talk about differences in the response of different types of microalgae to the adaptation process under increased illumination. Thus, *E. subsphaerica* is the culture least susceptible to an increase in the illumination intensity, i.e., the content of the major organic compounds (proteins, lipids, and carbohydrates) did not change significantly during all the experiments. The results for *Chlorella* can be considered similar, except for decreasing the number of proteins in the case of *C. vulgaris*, while protein content for *C. ellipsoidea* did not change significantly with increasing illumination. For the *G. pulchra* strain, an increase in the protein content entailed a decrease in the lipid content at the same illumination values (200 and 245 µmol quanta·m^−2^·s^−1^). In the case of *A. platensis*, a steady increase in the amount of carbohydrates was observed with an increase in illumination intensity. Also, an adapted express method was used to determine the viability of microalgae cells using a light microscope, based on cytochemical staining of living and dead cells with a lifetime dye methylene blue. This method made it possible to determine an insignificant proportion of dead cells at all light intensities, as well as morphological changes in the *A. platensis* trichomes at an illumination intensity of 245 µmol quanta·m^−2^·s^−1^.

Thus, the conditions for microalgae cultivation were optimized and the effectiveness of the method, which was used to adapt strains to high illumination intensity at elevated CO_2_ content, was demonstrated. 

## Figures and Tables

**Figure 1 plants-12-03876-f001:**
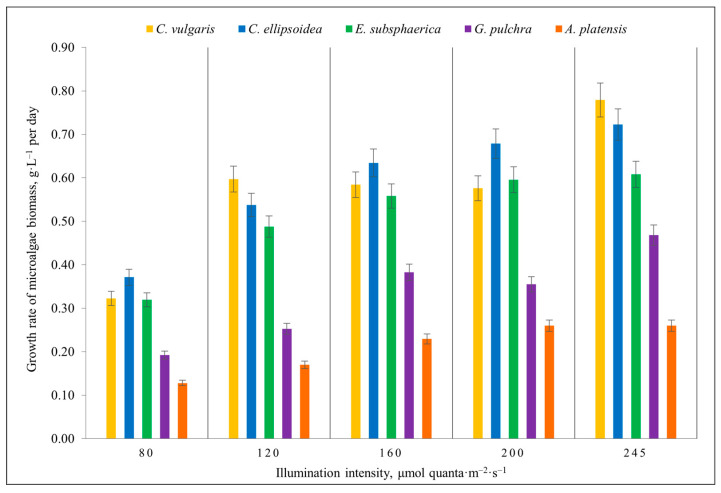
The growth rate of microalgae biomass (g·L^−1^·d^−1^) in the experiments with different illumination intensities. The culture duration is 12 days. The error bars represent the standard deviation.

**Figure 2 plants-12-03876-f002:**
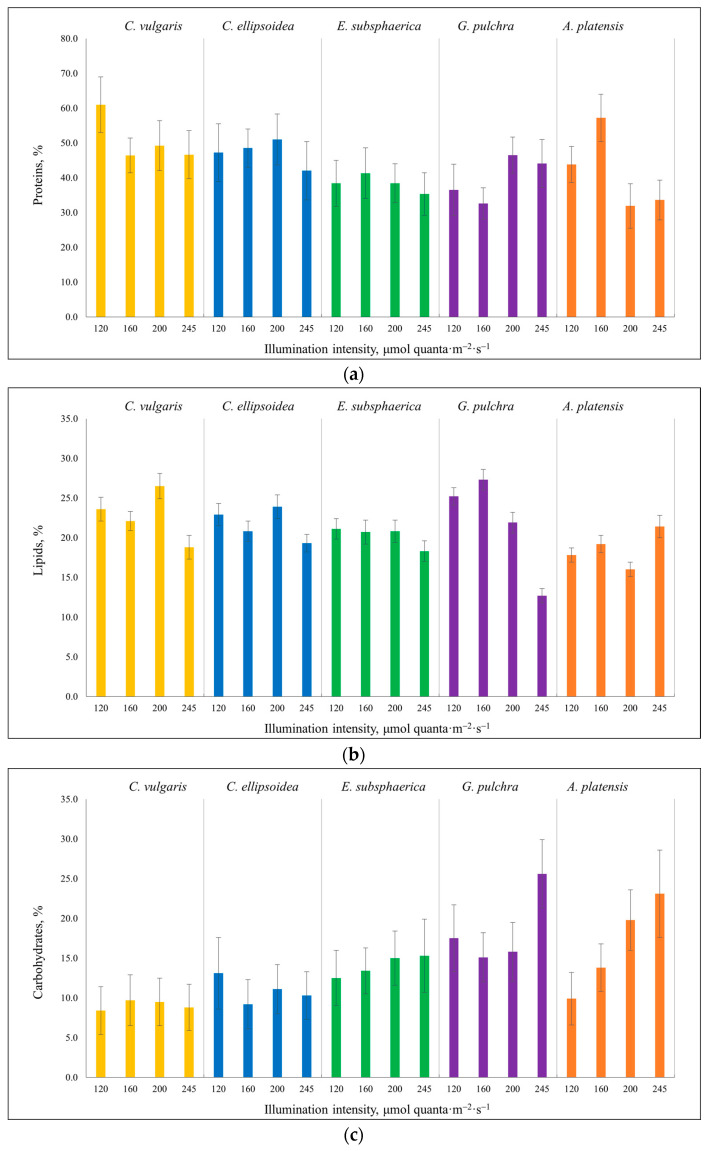
The results of biochemical analysis in the experiments with different illumination intensities: (**a**)—proteins, (**b**)—lipids, and (**c**)—carbohydrates. The culture duration is 12 days. The error bars represent the standard deviation.

**Figure 3 plants-12-03876-f003:**
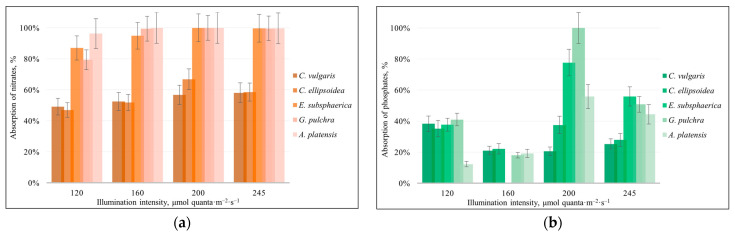
The consumption of nitrates (**a**) and phosphates (**b**) in the experiments with different illumination intensities. The error bars represent the standard deviation.

**Figure 4 plants-12-03876-f004:**
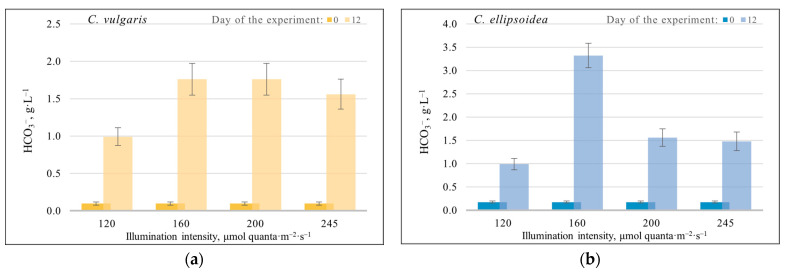
Change in the content of bicarbonates in the medium during cultivation of all strains in the experiments with different illumination intensities. (**a**)—*C. vulgaris*, (**b**)—*C. ellipsoidea*, (**c**)—*E. subsphaerica*, (**d**)—*G. pulchra*, and (**e**)—*A. platensis.* The error bars represent the standard deviation.

**Figure 5 plants-12-03876-f005:**
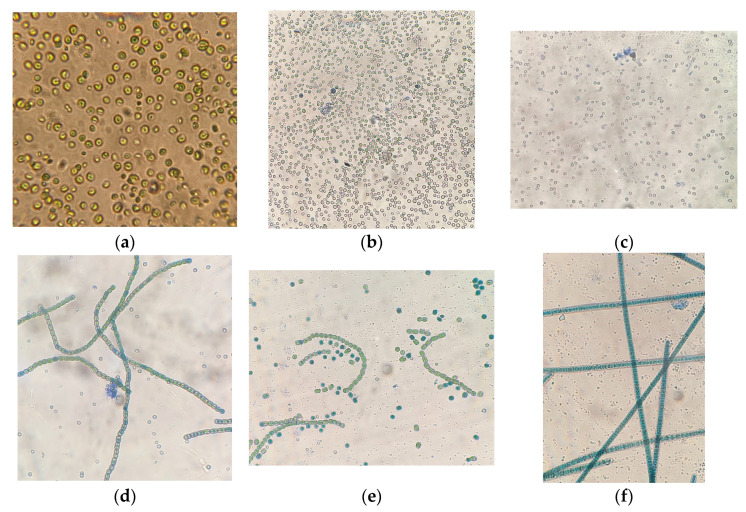
Cells of microalgae strains: (**a**) *C. vulgaris*, sample with staining, magnification ×1000. (**b**) *C. ellipsoidea*, (**c**) *E. subsphaerica*, (**d**,**e**) *G. pulchra*, and (**f**) *A. platensis*, samples with staining, magnification ×400, illumination intensity 120 μmol quanta·m^−2^·s^−1^.

**Figure 6 plants-12-03876-f006:**
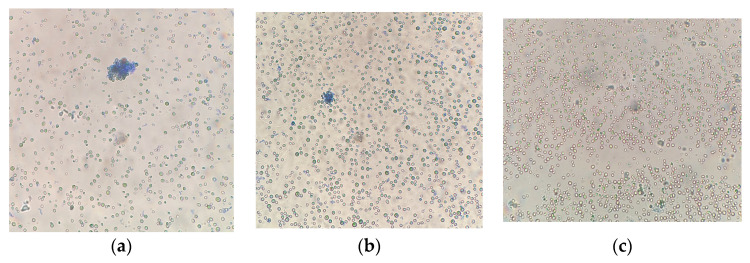
Cells of microalgae strains: (**a**,**d**) *C. vulgaris*, (**b**,**e**) *C. ellipsoidea*, and (**c**,**f**) *E. subsphaerica*. Samples with staining, magnification ×400. (**a**–**c**)—200 μmol quanta·m^−2^·s^−1^; (**d**–**f**)—245 μmol quanta·m^−2^·s^−1^.

**Figure 7 plants-12-03876-f007:**
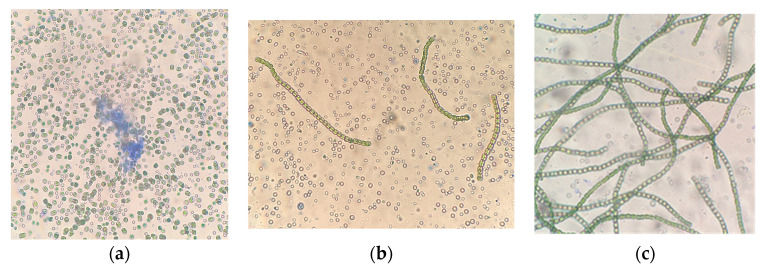
Cells of microalgae *G. pulchra*: samples with staining, magnification ×400. (**a**)—160 μmol quanta·m^−2^·s^−1^, (**b**)—200 μmol quanta·m^−2^·s^−1^, and (**c**)—245 μmol quanta·m^−2^·s^−1^.

**Figure 8 plants-12-03876-f008:**
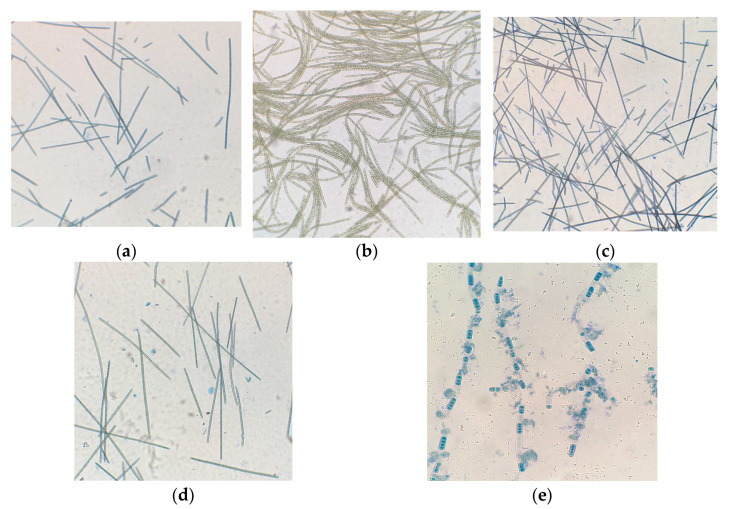
Cells of microalgae *A. platensis*; (**a**,**c**,**d**)—samples with staining, magnification ×100. (**b**)—sample without staining, magnification ×100; (**e**)—sample with staining, magnification ×400. (**a**)—160 μmol quanta·m^−2^·s^−1^, (**b**,**c**)—200 μmol quanta·m^−2^·s^−1^, and (**d**,**e**)—245 μmol quanta·m^−2^·s^−1^.

**Figure 9 plants-12-03876-f009:**
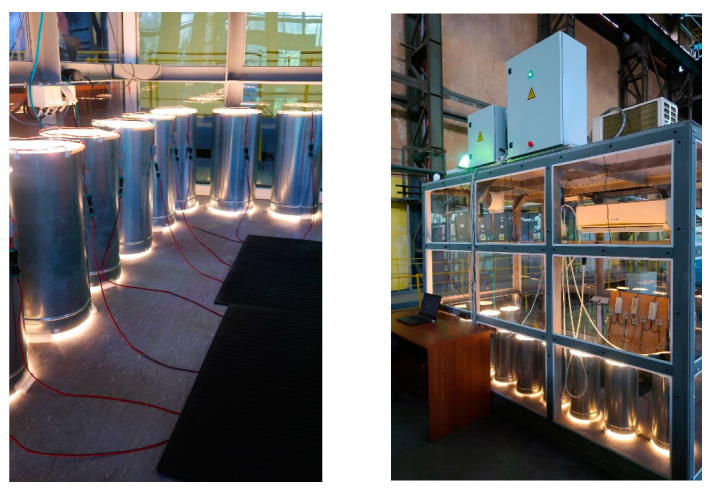
General view of the PBR system inside the AGC and the appearance of the AGC.

**Table 1 plants-12-03876-t001:** Comparison of the obtained biomass productivity with that reported in the literature for *C. vulgaris* and *A. platensis* strains.

Strains	*C. vulgaris*								*A. platensis*				
Biomass productivity, g·L^−1^·day^−1^	0.047	0.054	0.105	0.129	0.320	0.485	0.600	0.780	0.145	0.130	0.130	0.170	0.260
Reference	[16]	[16]	[40]	[41]	[11]	[42]	This study	This study	[41]	[43]	[11]	This study	This study
Illumination, μmol quanta·m^−2^·s^−1^, light:dark	55, 12:12	55, 12:12	150, 24:0	50, 12:12	74, 24:0	70, 24:0	120, 24:0	245, 24:0	50, 12:12	N.A., 24:0	74, 24:0	120, 24:0	200–245, 24:0
CO_2_ content, %	10	N.A.	10	5	6	2	6	6	5	5	6	6	6

**Table 2 plants-12-03876-t002:** Initial (0th day) and final (12th day) pH values in the experiments.

Strains	80 μmol quanta·m^−2^·s^−1^		120 μmol quanta·m^−2^·s^−1^		160 μmol quanta·m^−2^·s^−1^		200 μmol quanta·m^−2^·s^−1^		245 μmol quanta·m^−2^·s^−1^	
	Init. Value	Fin. Value	Init. Value	Fin. Value	Init. Value	Fin. Value	Init. Value	Fin. Value	Init. Value	Fin. Value
*C. vulgaris*	5.61 ± 0.00	6.87 ± 0.04	5.53 ± 0.00	8.32 ± 0.22	5.47 ± 0.00	8.08 ± 0.01	5.90 ± 0.00	8.03 ± 0.06	5.84 ± 0.01	8.49 ± 0.04
*C. ellipsoidea*	5.52 ± 0.01	6.84 ± 0.01	5.51 ± 0.01	8.32 ± 0.12	5.43 ± 0.01	8.13 ± 0.01	5.91 ± 0.01	8.08 ± 0.01	5.88 ± 0.00	8.49 ± 0.01
*E. subsphaerica*	8.83 ± 0.01	8.90 ± 0.22	8.53 ± 0.01	8.67 ± 0.16	8.33 ± 0.01	8.36 ± 0.08	8.89 ± 0.02	8.17 ± 0.04	8.79 ± 0.03	8.68 ± 0.10
*G. pulchra*	8.58 ± 0.00	8.77 ± 0.05	8.48 ± 0.00	8.55 ± 0.14	8.23 ± 0.00	7.79 ± 0.04	8.85 ± 0.01	8.06 ± 0.02	8.55 ± 0.02	8.59 ± 0.03
*A. platensis*	8.65 ± 0.01	8.79 ± 0.01	8.46 ± 0.01	8.81 ± 0.01	8.72 ± 0.01	8.73 ± 0.01	8.60 ± 0.01	8.80 ± 0.07	8.64 ± 0.00	8.84 ± 0.06

## Data Availability

Data are contained within the article.

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
