# Peer review of "Effects of Light Intensity on the Growth and Biochemical Composition in Various Microalgae Grown at High CO2 Concentrations"

_plants, 2023, doi:10.3390/plants12223876_

Round 1
Reviewer 1 Report
Comments and Suggestions for Authors
1- Arrange the keywords on the basis of alphabetic order.
2- Please, reduce the keywords into Maximum 5 keywords.
3- Each paragraph should start with new content, please, reduce the number of paragraphs into maximum 3 in your introduction and other parts of the manuscript
4- Results and Discussion, and conclusion parts are OK.
5- The abbreviation of journal s names should be written in references, for example, and the format of all references should be double checked.
Reviewer 2 Report
Comments and Suggestions for Authors
Please refer to the attached commented PDF for very minor suggestion for improvement.

Reviewer 3 Report
Comments and Suggestions for Authors
In this MS, the authors studied the influence of light intensity on the biomass growth and biochemical composition of 10 various microalgae strains. The optical density, pH, biochemical analysis and composition of culture medium were measured. Generally, the MS sounds meaningful. However, there are still some key issues need to improve and modify.
1. The English language of the MS need to improve.
2. Abstract. It should be modified. The background introduction, research purpose and also a short conclusion of this study are missing. Please consider to included them.
3. Keywords: the five names of the algae strains should be italic.
4. Introduction. I think this part is too long. Please consider to reduce the length of this part.
5. 2.1 the expression of the subtitles are not clear enough. Please consider to improve them in the whole MS.
6. Fig. 3 and Fig. 4. please add the statistical analysis results to the data in the two figures.
7. 3.5, please add the statistical analysis method to this part.
8. Fig. 5-8. Suggest to adjust the sizes of the sub-figure
9. “Summarizing the results of this study obtained under illumination by polychromatic white LEDs, it can be……”. The two sentences should be kept in the same tense, and it is recommended to use the simple past tense for both sentences.
10. “ The maximum productivity values (about 0.045h-1 were achieved at CO2=4% and illumination 100 μmol quanta·m−2·s-1.” Please note the usage specifications of punctuation marks.
11. “Microscopy of the strain G. pulchra grown at 160 μmol quanta·m-2·s-1 illumination showed that” Suggest changing 'showed' to 'shows’.
12. Note the use of punctuation in the annotation section of Figure 7.
13. “saturation is observed at 160 μmol quanta·m-2·s-1 illumination and growth rates were of approximately 0.25 and 0.59 g·L-1·d-1, respectively.”The two sentences before and after 'and' should be kept in the same tense.
14. Line 78,“growth rate almost twice as high - 0.140 g·L−1·d−1 ”, please regulate the use of dads.
15. Line 241, Table 2, it is suggested to adjust the layout to make the page orderly.
16. Line 297,“In [7], a similar trend was”,please standardize the citation and briefly mention the article or author information.
17. Line 379,“In the case of two strains Chlorella and a strain E. subsphaerica” has a syntax error, please add comma separation after strain.
18. There are many mistakes about the degrees Celsius units in the article, please check.
19. Line 523-570, as the conclusions part, i think it’s too long. Please consider to reduce the length of this part.
20. References, please recheck the mistakes in this part, such as, use the full name of the journal, or use the abbreviation?
Comments on the Quality of English LanguageThe English language of the MS need to improve.
Reviewer 4 Report
Comments and Suggestions for Authors
The MS showed the influence of light intensity on the biomass growth and biochemical composition of microalgae. Five typical microalgae strains were selected. The results give us some basal information for microalgae cultivation. in Figure 3, nitrogen is enough, but the accumulation of metabolite is high in Figure 2. In general, N limitaion might be one reason for this. You could discuss it. Scale bar in Figure 5-8 is needed.
Comments on the Quality of English Languageminor revision of language
Reviewer 5 Report
Comments and Suggestions for Authors
In this manuscript, the authors describe the growth of four green algae and a cyanobacterium at high CO2 (6%) and various photon flux densities. This work appears to be a continuation of Chunzhuk et al. (2023) Plants 12, 2470, in which the same species were tested at varying CO2 concentrations.
Overall, I have a lot of questions. I think that many of my questions are die to the fact that important information is missing from the manuscript. Some of the methods have no description or literature citation, and aspects of the experimental design are not entirely clear.
Major comments:
1) line 65 “For our work, it is necessary to note that the cells of both Chlorella sp. and C. vulgaris kept their normal morphologies after 15 day batch culture.”
Why did the authors point out that the normal morphology was maintained? Why is that relevant? Are there situations when morphology changes during a relatively brief batch culture? This should be explained. You cannot simply make an observation and expect the reader to piece together the logic.
2) Fig. 1, 2, legend – I suggest that additional details be added to the legend, most importantly, the culture duration should be mentioned. As well, what is the sample size (n)?
3) Fig. 3 – do the data of Fig. 3 indicate that N limitation occurred? And also that some P limitation may have occurred? Do these nutrient limitations potentially limit the yield? I am surprised that this was not addressed.
4) I am somewhat confused by the methylene blue staining. In the literature, there seems to be two schools of thought about methylene blue: 1) that methylene blue can enter ALL cells, and that healthy cells cause the blue colour to fade via enzymatic reduction of the dye (most of the yeast literature supports this interpretation), 2) that methylene blue is used in a dye exclusion assay, where live cells exclude the dye and dead cells do not. The authors should indicate what is being measured using methylene blue, and provide a reference.
Furthermore, if dead cells stain blue, then Fig. 5D has many dead cells. And Fig. 5F has ONLY dead cells. (As well, I think that there may be a problem in the figure legend, with 5E labelled as A. platensis; I suspect that it should be 5F.) As well, in Fig. 8, the stained samples of A. platensis are blue, regardless of the light levels. Does this mean that these cultures are largely dead? The cells in Fig. 8E are clearly dead (they appear to have ruptured), but are the non-ruptured cells on Fig. 8A, 8C and 8D also dead?
If the A. platensis cells are dead, how can there be a yield of protein, lipid and carbohydrate?
In summary, in my opinion, the staining work is very poorly explained. And the conclusions are unclear.
5) More about Fig. 5. The text describes some contamination of cultures with other algal species, and this is indicated in Fig. 5. Were the results from the cross-contaminated used in the calculation of results?
6) I suspect that the MQ-200 quantum sensor is not necessarily suitable for measuring PAR from LED light sources. In fact, the manufacturer even provides information about how to correct the sensor output for different light sources (https://www.apogeeinstruments.com/how-to-correct-for-spectral-errors-of-popular-light-sources-prior-to-june-1-2021/). Did the authors of this manuscript take into account the limitations of the sensor? Were corrections made?
7) Some of the figures have error bars, and some do not. For those figures with error bars, there is no indication what the bars represent (standard error, standard deviation, other?) and no indication of sample size (n). For the figures that do not have error bars, I am curious why there are none?
8) Table 2: The table heading should have more complete information. For example, it should indicate that that the final pH was measured after 15(?) days. There should also be an indication about sample size (n). As well, with respect to pH changes in the medium, the authors speculate about possible causes and also speculate about the buffering capacity of Zarrouks’s medium. However, there are no algae-free controls that are presented for comparison, e.g. what would have been the pH change in Zarrouk’s medium in the absence of Arthrospira?
9) In terms of sample size, I’m confused about how often the treatments were replicated. Was each treatment done in duplicate at the same time in the AGC? Were there any further replicates? If n=2, and the 2 treatments were done at the same time, is there the possibility of pseudoreplication?
10) line 234 – “In the case of Chlorella strains, a significant pH increase was observed in all experiments (Table 2). At the same time, many works [45–47] show that, when cultivating microalgae with bubbling air with high CO2 concentrations, pH decreases, that is, acidification. Our results might be explained by the fact that Chlorella was grown on a Tamiya nutrient medium, which is unstable, since during the growth of microalgae, the physiological absorption of NO3- anions prevails compared to Na+ cations, which accumulate in the nutrient medium, which leads to an increase in pH.”
This requires additional explanation. What is unstable about Tamiya’s medium? Are there literature references about the “instability”? As well, all algal culture media have numerous inorganic salts, including Na salts. What is the reason that Tamiya’s medium is especially susceptible to a pH increase compared to other standard media?
11) line 321 – “However, for the G. pulchra, at the illumination of 200 μmol quanta·m-2·s-1, the absorption of phosphates reaches almost 100%, and at 245 μmol quanta·m-2·s-1 decreases to 50%. It can be explained by the fact that the studied microalgae, with a deficit of mineral nutrition, are characterized by a transition to mixotrophic growth. Since the culture is not axenic, heterotrophic bacteria are present in it, which provide additional nutrition to microalgae with mineral and low-molecular-weight organic nitrogen and phosphorus.”
I find it very difficult to accept this explanation. No evidence is provided, nor are any literature citations provided. As well, the reasoning does not hold up, in my opinion. Where are the heterotrophic microbes getting their N and P, if not from the medium? If the heterotrophs are using medium N and P, then the N and P levels should decrease. As well, why would the algal cells use up all of the N and P at one light level, but not at another, slightly higher light level? Does higher light promote mixotrophy for some reason? The reasoning does not hold up.
12) line 351 – “A decrease in HCO3- may be due to the alkalinization of the medium, since an increase in pH may lead to the reduced availability of HCO3- for microalgae.”
If it is assumed that the CO2 level was saturating in the medium (it is not clear if that is the case; it might be useful to know that), then an increase in pH would lead to an increase in the equilibrium bicarbonate concentration. Furthermore, many microalgae will use CO2 quite happily, especially if the CO2 is easily available. They might not using bicarbonate at all.
Minor comments:
1) line 429 - “resistant consortium of microalgae/cyanobacteria Arthrospira platensis”
It might be worthwhile to point out that all of the other strains used in this work are green algae (and thus eukaryotes), while Arthrospira is a cyanobacterium (and prokaryotic). As well, it is not clear what is meant by a “resistant consortium”.
2) line 123 - Why is "oleaginous" italicized?
3) line 131 – “840.56 mg·L−1·d-1"
All the other productivity rates were expressed in units of g L-1 d-1.
Comments on the Quality of English Language
Could use a small amount of improvement, for clarity.
Round 2
Reviewer 3 Report
Comments and Suggestions for Authors
In this revision, the MS has been improved much. But there are still some issues need to improve and modify.
1. The modified authors added in the author list has obvious format error. Please check.
2. Keywords, ‘illumination’ seems not suitable, please consider.
3. Introduction. I think this part is still too long. Especially for the second paragraph. Please consider to reduce the length of this part.
4. Please adjust the position of Fig.1.
5. Fig. 5-8. consider to adjust the sizes of the sub-figure.
6. Line 472-514, as the conclusions part, I think it’s still too long. Please consider to reduce the length of this part.
Comments on the Quality of English LanguageMinor editing of English language is still required.
Reviewer 5 Report
Comments and Suggestions for Authors
The authors have addressed my comments on the earlier version. Thank you.
Author Response
Thank you!